# Manganese Intoxication Induced by Total Parenteral Nutrition in the Intensive Care Unit: A Case Report

**DOI:** 10.3390/diagnostics15111346

**Published:** 2025-05-27

**Authors:** Victoria Seijas-Martínez-Echevarría, Rita Martínez-Manzanal, Ester Mena-Pérez, Pilar Nuñez-Valentín, Guadalupe Ruiz-Martin

**Affiliations:** 1Facultad de Ciencias Biomédicas y de la Salud, Universidad Alfonso X el Sabio (UAX), Avenida de la Universidad 1, 28691 Villanueva de la Cañada, Madrid, Spain; ester.mena@salud.madrid.org (E.M.-P.); guadalupe.ruiz@salud.madrid.org (G.R.-M.); 2Clinical Laboratory, Hospital Universitario Severo Ochoa, Avenida de Orellana s/n, 28911 Leganés, Madrid, Spain; manzanalrita@hotmail.com; 3Radiology Service, Hospital Universitario Severo Ochoa, Avenida de Orellana s/n, 28911 Leganés, Madrid, Spain; pnunezv@salud.madrid.org

**Keywords:** nutrition, neurotoxicity, parenteral nutrition, trace elements, manganese

## Abstract

**Background:** Manganese (Mn) is an essential trace element for humans. It has been recognized as a potential occupational toxin, but its danger as a toxin in patients under parenteral nutrition is often forgotten. **Case Presentation:** A 73-year-old man was logged for 210 days in the intensive care unit (ICU), receiving parenteral nutrition (PN) for a month, and was then transferred, first, to the internal medicine ward and, then, to the rehabilitation hospital, and 223 days after discharge from the ICU, he had current disease, chorea-type movements in the head and neck, and left hemibody. Diagnostic tests: The magnetic resonance imaging findings suggested manganese deposits, with a total blood manganese concentration of 34 µg·L^−1^ (reference range: less than 13 µg·L^−1^). Discussion: Abnormal movements can be caused by manganese poisoning due to parenteral nutrition and are associated with liver failure in the ICU. Our patient showed toxic Mn concentrations in whole blood after 31 days of receiving 300 μg·d^−1^ of Mn in PN, a shorter duration than typically reported. Neurotoxicity was observed several months later (223 days). Factors such as liver dysfunction and iron deficiency can modulate neurotoxicity. Age may also be a susceptibility factor due to increased expression of Mn transport proteins. Magnetic resonance imaging (MRI) intensity in the globus pallidus is useful for detecting brain Mn accumulation, but it is not feasible for routine clinical practice. **Conclusions:** In this case, choreiform movements were attributed to manganese (Mn) accumulation in the basal ganglia. It is essential to monitor patients receiving parenteral nutrition (PN) solutions containing Mn, especially in those who have biomarkers of susceptibility, even if they have not yet shown neurological signs, and routinely measure whole-blood Mn concentrations, iron levels, age, and liver function. If Mn intoxication is suspected, a brain MRI examination should be conducted.

## 1. Introduction

Manganese (Mn) is an essential trace element for human life, and its various metabolic functions require its presence as a nutrient. It is involved in bone formation, amino acid metabolism, energy metabolism, and as an antioxidant, among other essential functions [1,2].

Despite the classification of Mn as an essential trace element, there is scant evidence of clinically relevant Mn deficiency in humans [3]. Such deficiencies are exceedingly rare, with clinical symptoms including weight loss and coagulation issues [4]. Studies show that Mn deficiency may be important in carcinogenesis because it causes the activation of p53 [5].

Typically, human disease associated with Mn abnormalities is found in excess, not deficiency, of Mn. There are genetic pathologies known as familial hypermanganesemia with dystonia or familial neurotoxicity due to accumulation of Mn, of which two types are known: hypermanganesemia combined with polycythemia, cirrhosis, and dystonia, type 1, and hypermanganesemia with dystonia type 2, due to mutations in the SLC30A10 and SLC39A14 genes [6,7].

Human exposure to Mn primarily occurs in occupational environments, such as ferrous alloy production, mining, foundries, battery manufacturing, and welding industries. Considering the extensive industrial use, it is important to consider occupational poisoning, known as manganism, whose symptoms resemble those of Parkinson’s disease [8]. Beyond occupational exposure, Mn-contaminated food and drinking water represent a primary source of non-occupational Mn toxicity [9].

This case report is based on another route of Mn poisoning, parenteral nutrition (PN). Although PN was introduced into medical practice in the 1960s [10], the iatrogenic risk of PN-associated Mn-induced neurotoxicity was only recognized in 1990 [11], when the case of a 32-year-old woman receiving Mn daily was reported. After 4 months of Mn supplementation, the patient developed extrapyramidal signs. Since the first report of Mn-induced neurotoxicity, other cases of Parkinsonian-like symptoms associated with Mn exposure from parenteral mixtures have been reported [12,13,14,15,16,17,18,19,20,21,22,23,24,25,26].

Only a small fraction of dietary Mn is absorbed, and a fraction of it is retained after biliary elimination. However, in PN, these homeostatic mechanisms are disrupted, particularly if there is simultaneous liver dysfunction. Manganese toxicity from parenteral nutrition can result from excessive accumulation in the brain, reduced excretion in liver disease, oxidative stress, and disruption of neurotransmitter function, leading to neurological symptoms.

It would be interesting to establish more complex biomarker panels. Exposure biomarkers, such as the measurement of Mn in whole blood (WB-Mn), should be determined according to guidelines if markers of susceptibility to poisoning are present, and magnetic resonance imaging (MRI) scans are not routinely employed as a screening test.

## 2. Case Report

A 73-year-old man was referred to neurology consultation from a long-stay residence for evaluation of abnormal movements.

### 2.1. Personal Background

The patient had no known drug allergies. The patient presented with type 2 diabetes mellitus, treated with oral antidiabetic medications, with a last recorded glycated hemoglobin (HbA1c) level of 4.6% (27 mmol·mol^−1^). The usual treatment was 20 mg of omeprazole, 1-0-0, and 1 mg of repaglinide, 0-1-0.

The summary of the clinical examination described no history of hypertension and no toxic habits. The patient was evaluated by pneumology due to dyspnea. Thoracic computed tomography (CT) scans showed parenchymal changes suggestive of interstitial neuropathy and non-idiopathic pulmonary fibrosis, along with emphysema and bronchiectasis. Given the patient’s clinical condition, no further complementary studies or specific treatments were pursued. The patient’s surgical history included hemorrhoid surgery, cholecystectomy, and cataract surgery in the left eye. The patient’s baseline functional status was as follows: The patient lived in a family home, ambulated with a cane, and had mild-to-moderate cognitive impairment.

The patient presented with gastric adenocarcinoma, presenting as a large nodular mass in the gastric body, staged as T3NXM0, and was treated with chemotherapy (three cycles of the epirubicin–cisplatin–fluorouracil (ECF) scheme) and subtotal gastrectomy, Billroth II surgery. Five days after surgery, the patient presented with hypotension (70/30 mm Hg), tachycardia (heart rate: 120 bpm), and tachypnea. The radiographic findings were suggestive of acute pulmonary edema. He was reoperated after being found to have supramesocolic peritonitis due to dehiscence of anastomosis in the gastric side of the previous jejunostomy. The patient presented with candidemia in the ICU associated with catheter infection, bacteremia by *S. aureus MS*, recurrent pneumonia, and septic shock with multiple organ failure, and remained logged for 210 days in the ICU, receiving PN for 31 days. The patient was delayed in resuming consciousness after the withdrawal of sedation, and later, he needed neuroleptics and benzodiazepines.

Serum glucose, creatinine, calcium, copper, zinc, and phosphorus concentrations and serum TSH (thyroid-stimulating hormone) levels were normal.

The patient showed iron-deficiency anemia and data indicating altered hepatic excretion, as shown by the biochemical variables in Table 1.

After discharge from the ICU to the internal medicine ward, the patient had severe generalized atrophy and paresis. Several months later, he was moved to a long-stay residence for the rehabilitation of severe critical illness polyneuropathy.

We used the Pierson, Bradford Hill, and Newcastle–Ottawa tool to determine the degree of bias and the quality of evidence of the case description and found that it duly met the selection, verification, causality, and reporting criteria [27].

### 2.2. Current Disease

Doctors at the long-stay residence reported chorea-type movements in the head and neck and left hemibody that increased and were unrelieved with anticholinergics or benzodiazepines. The movements made walking impossible. The patient appeared conscious, attentive, and oriented; his speech was normal.

Regarding the cranial nerves (CNs), the patient exhibited a normal fundus oculi (FO) and normal extraocular movements (EOMs). Regarding oculomotor reflex (OR), the patient showed a normal motor and consensual response. No facial asymmetries were observed. The lower cranial nerves were normal. The motor examination showed a strength of 4+/5 in both the upper and lower limbs, with absent deep tendon reflexes (DTRs). Pain and touch sensitivity were preserved. No dysmetria or adiadochokinesia was observed. Walking was impossible. He was admitted to the neurology ward for study. 

### 2.3. Additional Tests

*Syphilis*, *Borrelia*, *Brucella*, and HIV and HBV serology were negative.

Onconeural antibodies (anti-Hu-, anti-Yo-, anti-Ri-, and anti-CMV2) were negative.

During admission, treatment with haloperidol and clonazepam was started, with good tolerance and significant improvement in his movements, allowing him to walk with a walker.

T1-weighted magnetic resonance imaging (MRI) revealed symmetrical high-intensity lesions in the globus pallidus. There were no ischemic or demyelinating lesions (Figure 1 and Figure 2).

A paraneoplastic origin of the movement was dismissed, basal ganglia metastasis was ruled out, and no metabolic disorders (different from the previous diabetes mellitus) were found that could justify the choreiform movements. No ischemic or demyelinating lesions were observed. The only significant finding was observed in the MRI, which revealed T1 hyperintensity in the basal ganglia, a finding suggestive of metal deposition. The Mn (II) ion has five unpaired electrons in the 3d orbit, causing the shortening of the T1 relaxation time and an increase in the signal intensity in T1-weighted MRI. Signs of supra- and infratentorial atrophy were observed with the enlargement of sulci, cisterns, and the ventricular system, correlating with age.

Upon reviewing the patient’s medical history, it was noted that during the ICU stay, the patient developed liver failure with a cholestatic pattern, secondary to multiorgan failure associated with septic shock, parenteral nutrition, and pharmacological treatment. Further analysis revealed that the patient was receiving parenteral nutrition during the period in which liver failure occurred.

An evaluation of the parenteral nutrition formula showed the Mn content to be 300 µg of Mn·day^−1^, nominal for 31 days. In 2012, the ASPEN revised its recommended manganese dose from 60–100 µg·d^−1^ to a lower dose of 55 µg·day^−1^, a recommendation increasingly supported by experts [28,29]. Considering these recommendations, the patient’s manganese intake appeared to be elevated and potentially toxic. Despite the fact that the exposure period lasted for only 31 days, this excessive exposure likely contributed to manganese accumulation, as evidenced by the MRI findings and the subsequent development of chorea. The presence of liver dysfunction, which may have impaired manganese clearance, further supported the likelihood of manganese toxicity as the underlying cause of the patient’s neurological symptoms. This diagnosis was confirmed by the quantification of manganese levels using graphite furnace atomic absorption spectrophotometry. Mn determination in whole blood by GFAAS provided a total of 34 µg·L^−1^, higher than the normal upper limit (13 µg·L^−1^).

After 12 months, the whole-blood Mn concentration decreased to 8.1 µg·L^−1^, and T1-weighted MRI no longer showed high-intensity lesions in the globus pallidus (Figure 3 and Figure 4).

One year later, treatment with haloperidol could not be stopped due to the worsening of his involuntary movements. Upon neurological examination, there was no rigidity or bradykinesia. Choreiform movements did not prevent the maintenance of the sitting posture, nor was there any impact on walking. He did not tiptoe and did not present tremor.

## 3. Discussion

The recommended doses of Mn in PN have a wide range that can vary from 10–50 µg·d^−1^ to values of 300 µg·d^−1^. In 2004, the ASPEN (American Society of Parenteral and Enteral Nutrition) modified the ADA guidelines on the recommendations of Mn in PN, and the daily intake of Mn for adults was reduced to 60–100 μg·d^−1^ [28]. Most of the case reports of Mn intoxication were in adults receiving more than 500 μg·d^−1^ of long-term parenteral Mn. In our patient, toxic concentrations of Mn were measured in whole blood after the onset of Parkinsonian movements. These were reported months after having received 300 µg·d^−1^ of Mn in the PN for 31 days, a period lower than that usually reported [30], although there are cases with 14–15 days of PN with higher doses of Mn [14,16,17,18,20].

In addition to a parenteral dose of Mn of greater than 500 micrograms per day, the duration of PN has also been described as a possible modulating factor of neurotoxicity. In vivo studies suggest that the active dopamine transporter (DAT) plays a key role in the accumulation of Mn in the striatum. Inhibition of DAT function in rats attenuates this accumulation of Mn during chronic exposure [31,32].

This patient had several factors that should be considered. On the one hand, he had previously undergone gallbladder surgery, and during his admission to the ICU, he presented with liver failure. Alterations in this Mn elimination pathway should be considered in the assessment of Mn toxicity [33,34,35]. Iron (Fe) deficiency can increase brain Mn levels [36], as it is associated with high concentrations of serum transferrin receptors, which are transporters present at the blood–brain barrier that mediate the brain influx of both Fe (II) and Mn (the predominant reduced form in the human body) [37]. Transport of Mn (II) within the central nervous system (CNS) is also carried out by a family of proteins known as natural resistance-associated macrophage proteins (NRAMPs) [36,38]. These proteins are expressed more with age [39]. This complements the hypothesis that age may be another individual factor of susceptibility to Mn [40].

Our laboratory has graphite furnace atomic absorption spectrometry (GFAAS) (PinAAcle 900-Z Perkin-Elmer, Waltham, MA, USA), with Zeeman correction, and has validated methods for the determination of WB-Mn [41]. The whole-blood samples were collected in trace-metal-free tubes to prevent contamination. The samples were diluted with a matrix modifier (2% palladium nitrate) to break down proteins and improve atomization at 2600 °C. During atomization, Mn atoms absorbed light from the Mn-specific lamp at 279.5 nm, and the absorbance was measured. The absorbance was directly proportional to the Mn concentrations in the sample. Certified reference materials were measured to ensure accuracy.

For the evaluation of toxicity in cases such as the present one, the use of susceptibility markers, such as age, the presence of cholestasis, liver failure, or iron deficiency, can help to complete the diagnostic orientation [42]. This leads to a more intensive search for Mn poisoning in patients with those criteria (advanced age, iron deficiency, and liver failure). Regarding exposure markers, although WB Mn analysis might not be precise enough for individual assessment, it is the preferred screening method [43]. In a large review published in 2022 [9] on biomarkers of environmental exposure, a variability was observed in the reference values of Mn in whole blood published in unexposed individuals [44]. The WB-Mn here measured at the onset of neurotoxicity symptoms (34 μg·L^−1^), measured several months after the end of PN, represented a concentration higher than that of any unexposed individual, higher than our reference value for unexposed people (13 μg·L^−1^). Although the concentration may not appear significantly divergent from the reference values reported by other authors [45], it was notably higher than those documented in the northern region of Spain for a non-exposed population [46]. Furthermore, our reference values align with those published in other countries, such as the United States (4–14 µg·L^−1^) [47]. Most importantly, considering the elapsed time since exposure, this concentration should be regarded as toxic.

The possible early clinical manifestations of manganism, such as neurological symptoms (a reduced response speed, irritability, mood changes, and compulsive behaviors), psychiatric symptoms (psychosis, mood swings, depression, and compulsiveness), and physical symptoms (weakness and lethargy), were not identified in this case since their long stay in intensive care could have masked them.

At the time of the end of PN, we assumed that the concentration of Mn was higher, but it was not measured because neurotoxicity manifested itself late. A year later, when the hyperintensity disappeared in the MRI, the WB-Mn level had dropped to 8 μg·L^−1^. MRI intensity in the globus pallidus is a useful means for the detection of brain Mn accumulation [48], but repeated performance during PN is not feasible in routine clinical practice, so its indication refers more to a differential diagnosis in cases of the appearance of neurotoxicity symptoms, rather than as a search for Mn poisoning.

The use of other biomarkers, such as nails or hair, is not advisable due to their enormous variability and the risk of contamination [49,50].

Regarding the treatment of Mn poisoning, the most accessible form of treatment for manganism is to remove the patient from the source of exposure, whether occupational, environmental, or iatrogenic.

Chelation therapy for manganism involves the use of CaNa_2_ EDTA (EDTA) and para-amino salicylic acid (PAS). The use of EDTA [51] has been shown to effectively increase the urinary concentration of Mn and decrease its blood levels. These findings did not coincide with the observed decrease in clinical toxicity due to the chronic nature of toxicity and its incomplete reversibility. However, EDTA prevents more Mn from crossing the blood–brain barrier, disabling its ability to enter the CNS and exert its toxic effects. Unfortunately, EDTA has a low bioavailability to the brain parenchyma and does not appear to be able to fully reverse the toxic effects of Mn. Therefore, in patients with chronic toxicity or advanced manganism, chelation therapy is not likely to reverse the clinically significant deterioration. In addition, a rebound phenomenon often occurs, with side effects such as the loss of zinc (Zn), iron (Fe), and calcium (Ca), or its redistribution in tissues, complexing agent syndrome, and renal tubular necrosis, among others.

Another chelating molecule, para-amino salicylic acid (PAS), is authorized for use as an anti-tuberculosis drug, but has demonstrated clinical benefits in patients with manganism [52].

In addition to chelation therapy, iron supplementation has been shown to improve neurological symptoms compared with a treatment group that received chelation alone [52].

The patient in question had stopped receiving PN, and chelation was ruled out due to his comorbidity.

## 4. Conclusions

In the present case, the justification for the choreiform movement pertained to the accumulation of Mn in the basal ganglia during PN administered months ago.

It is imperative to monitor patients receiving PN solutions containing Mn for neurological signs and to routinely measure WB-Mn, assess for iron deficiency, consider the patient’s age, and conduct liver function tests. In the presence of biomarkers indicating susceptibility, it is recommended to consider performing a brain MRI examination, discontinue manganese intake, and evaluate the potential use of chelation therapy.

## Figures and Tables

**Figure 1 diagnostics-15-01346-f001:**
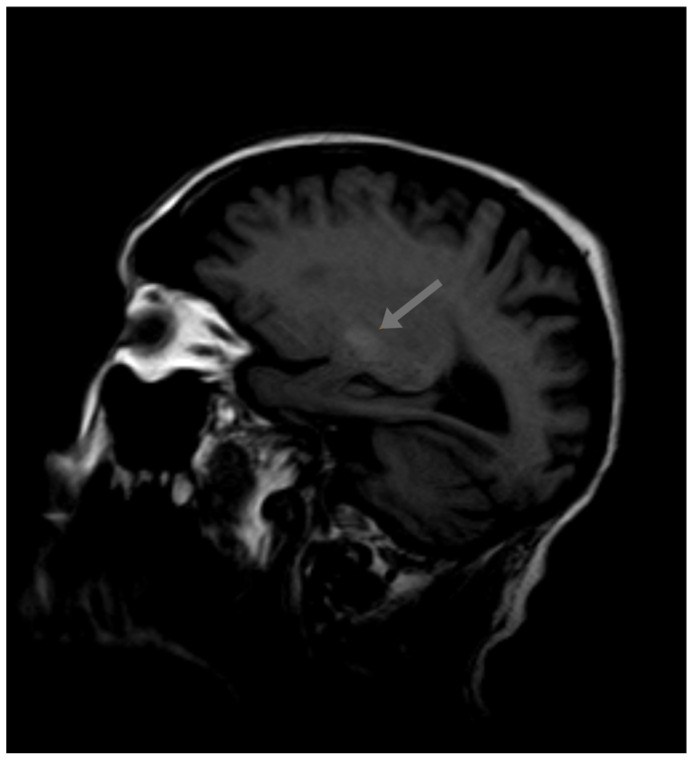
Initial MRI.

**Figure 2 diagnostics-15-01346-f002:**
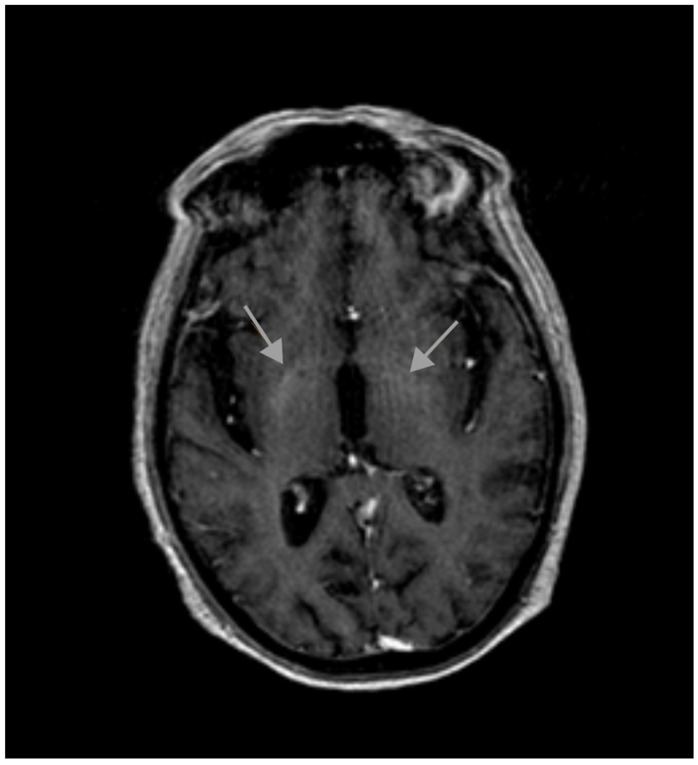
Initial MRI.

**Figure 3 diagnostics-15-01346-f003:**
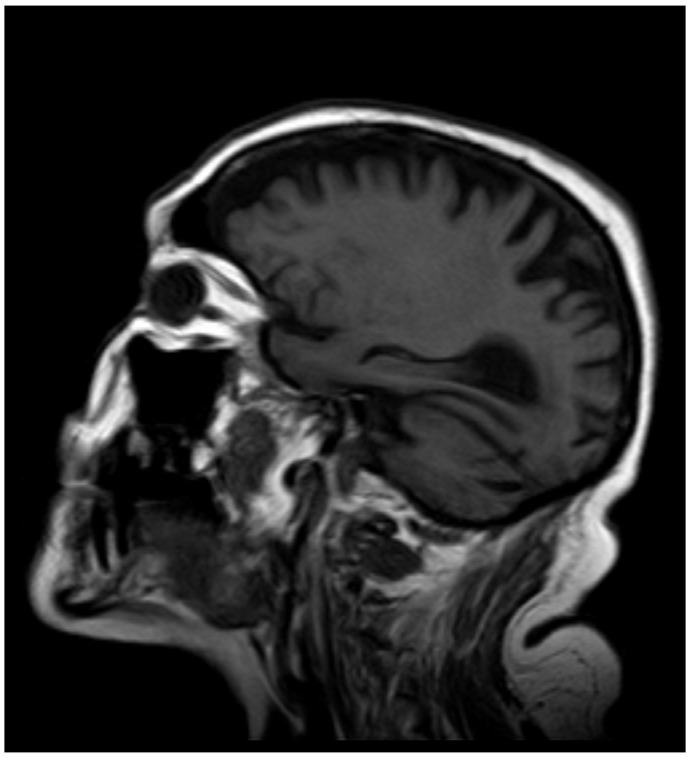
MRI after 12 months.

**Figure 4 diagnostics-15-01346-f004:**
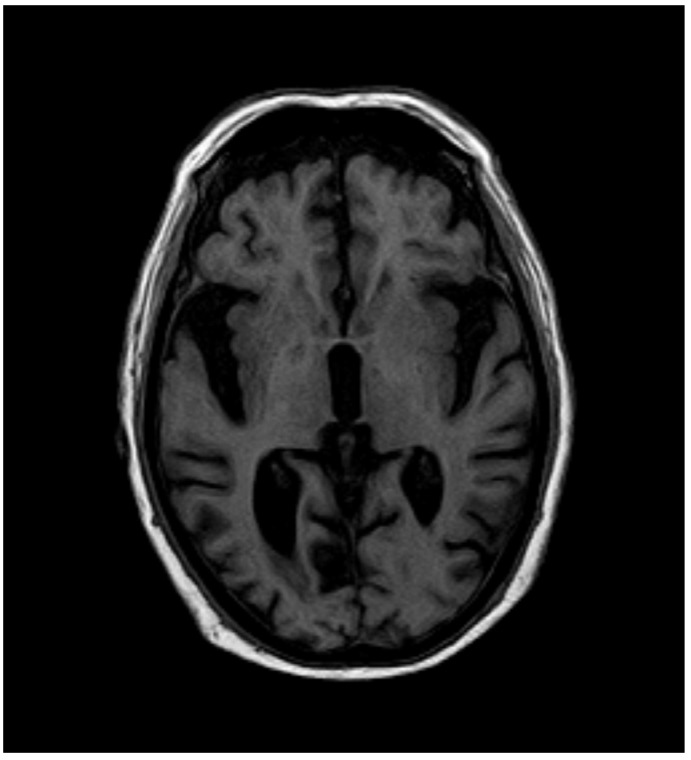
MRI after 12 months.

**Table 1 diagnostics-15-01346-t001:** Relevant biochemical variables.

	Hb (g·dL^−1^)	Iron (µg·dL^−1^)	GGT (U·L^−1^)	Bil (mg·dL^−1^)	AST (U/L^−1^)
At the beginning of PN	9.1	14	173	2.5	0.6
At the end of PN	9.6	48	396	0.6	69
Reference range	13–18	57–182	11–49	<1.1	<37

## Data Availability

This article, being a case report and review, does not contain any primary data for sharing. The data discussed were derived from previously published studies and the patient’s medical records.

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
