# Peer review of "Manganese Intoxication Induced by Total Parenteral Nutrition in the Intensive Care Unit: A Case Report"

_diagnostics, 2025, doi:10.3390/diagnostics15111346_

Round 1
Reviewer 1 Report (Previous Reviewer 2)
Comments and Suggestions for Authors
In my opinion, the manuscript is now significantly improved and authors have done adequate modifications besides clarifying all my major concerns. It may now be accepted for publication
Reviewer 2 Report (Previous Reviewer 1)
Comments and Suggestions for Authors
the manuscript can be accepted
This manuscript is a resubmission of an earlier submission. The following is a list of the peer review reports and author responses from that submission.
Round 1
Reviewer 1 Report
Comments and Suggestions for Authors
- In the introduction, the mechanism of Manganese poisoning induced by total parenteral nutrition in the ICU is unclear.
- According to Dystonia and movement disorders have been reported in patients receiving total parenteral nutrition (TPN) for extended periods, please insert evidences.
- Add future directions to the discussion section.
- Some grammatical improvements could enhance clarity, particularly in the Abstract and Introduction.
Some grammatical improvements could enhance clarity, particularly in the Abstract and Introduction
Reviewer 2 Report
Comments and Suggestions for Authors
This manuscript provide a short case report of 73-year-old man admitted in ICU with liver failure. Authors correlated high manganese level with TPN solutions with > 0.1 mg/ day leading to Chorea - type movement. In my opinion the study raises more questions than it answer besides there are number of serious shortcomings. Some of my major concerns include -
- My first major concerns is about the length of the manuscript. A short case report has been unnecessarily been dragged and most of the text is not required or not related to the study. I would like to specifically cite Introduction and Conclusion which need to be shortened considerably. In my opinion the manuscript needs to be shortened considerably (almost ¾ of the text needs to be deleted) before any possible consideration.
- Authors provided inadequate details about the patient’s history including providing details about the clinical picture and biochemical variables. Such information in a tabular form could have been useful. Why no picture/ figure were included showing manganese contents using MRI.
- Authors suggest that whole-blood manganese levels are commonly used as exposure markers and suggest instead MRI scan as necessary for confirmation. Authors further suggest that Mn supplementation should be used with caution. My concerns is what is new (novelty) in such conclusion. There are numerous contradictory statements throughout the text.
- Why chelation therapy was not tried and authors suggesting, instead, to stop supplementation and wait for more than a year for manganese level to reduce?
Reviewer 3 Report
Comments and Suggestions for Authors
The reviewed paper reports a case of an oncological patient with several co-morbidities and postoperative complications (septic shock with multiple organ failure) who received total parenteral nutrition for 2 months and developed motoric dysfunction (chorea-type, parkinsonian-like movements) which was attributed to elevated manganese concentration.
It is generally well known from previous studies that hypermagnesemia is a frequent feature in patients receiving parenteral nutrition. Patients receiving total parenteral nutrition often present hypermagnesemia caused by cholestatic liver disease and thereby decreased Mn biliary excretion. Clinical reports indicate such a connection dating back to the 90s.
The article's subtitle (“case report and review”) is misleading because the manuscript does not provide a thorough review of previously published studies nor does it summarize the findings of previous reports. Moreover, some of the content has not been supported by any citations.
So, it is difficult to find any novelty in the submitted work, and the scientific and practical utility of the presented case is limited.
The structure of the manuscript is also questionable and requires reconstruction. The discussion section begins with a sentence that is essentially a conclusion: “Parenteral exposure to Mn with impaired excretory gallbladder has been the cause of the poisoning, suspected by the radiological images.” The reviewed paper generally lacks proper discussion – some elements of the discussion are scattered throughout the case description section. Confirmation of actual manganese ‘poisoning’ requires careful differentiation of other potential causes of the observed disorders. First of all, a more in-depth discussion is needed on the reference range of toxic manganese concentrations. The reported concentration of 34 μg·L-1 is close to reference values reported in some population studies (e.g. Freire et al. 2015, Baj et al. 2023). Moreover, increased signal intensity in MRI may be due to the deposition of other paramagnetic substances. Neurotoxic effects are generally poorly correlated with elevated Mn levels and brain manganese accumulation did not always result in neurologic disorders. Chorea-type walking difficulties are considered late-onset symptoms of hypermagnesemia (as well as speech problems and increased reflexes which were absent in the presented case). The authors should also consider and discuss early clinical manifestations that give way to late-stage manganism.
The authors should also address other questions that arise while reading their paper:
Lines 75-76: the following statement seems incorrect: “The literature lists (6) several cases of manganese toxicity linked to the use of parenteral solutions containing manganese.” In the article by Reimund at. al (reference no. 6) among 21 participants only one patient had mild extrapyramidal symptoms only disclosed at careful neurologic physical examination.
Line 161: “and no metabolic disorders were found” – compare line 106 statement.
Lines 162-164: “The only significant finding was observed on magnetic resonance imaging (MRI), which revealed T1 hyperintensity in the basal ganglia, a finding suggestive of metal deposition.” – it would be very desirable to provide an appropriate MR scan illustrating this observation
Were there any repeated MRI scans before or after hospitalization in the ICU?
Were iron levels checked during treatment? Mn and Fe compete for absorption in the body and dietary Fe deficiency can lead to excess absorption of Mn (which is consistent with the statement in lines 84-85).
Has the possibility of determining Mn in an alternative material, e.g. hair or nails, been considered?
Other issues:
Lines 45-51: “Epidemiological research has indicated that…” – this paragraph lacks any reference.
Page 119: “Bill Roth Il surgery” – misnaming of Billroth's operation II, named by the German surgeon.
Page 8: Some of the references lack journal titles.
